# 4D nucleome equation predicts gene expression controlled by long-range enhancer-promoter interaction

**Zihao Wang**[1,2], **Songhao Luo**[1,2], **Zhenquan Zhang**[1,2], **Tianshou Zhou**[1,2]*, **Jiajun Zhang**[1,2]*

**1** Guangdong Province Key Laboratory of Computational, Sun Yat-sen University, Guangzhou, People's Republic of China, **2** School of Mathematics, Sun Yat-Sen University, Guangzhou, People's Republic of China

\* mcszhtsh@mail.sysu.edu.cn (TZ); zhjiajun@mail.sysu.edu.cn (JZ)

**Data Availability Statement:** The source code and data used to produce the results and analyses presented in this manuscript are available from

## Abstract

Recent experimental evidence strongly supports that three-dimensional (3D) long-range enhancer-promoter (E-P) interactions have important influences on gene-expression dynamics, but it is unclear how the interaction information is translated into gene expression over time (4D). To address this question, we developed a general theoretical framework (named as a 4D nucleome equation), which integrates E-P interactions on chromatin and biochemical reactions of gene transcription. With this equation, we first present the distribution of mRNA counts as a function of the E-P genomic distance and then reveal a power-law scaling of the expression level in this distance. Interestingly, we find that long-range E-P interactions can induce bimodal and trimodal mRNA distributions. The 4D nucleome equation also allows for model selection and parameter inference. When this equation is applied to the mouse embryonic stem cell smRNA-FISH data and the E-P genomic-distance data, the predicted E-P contact probability and mRNA distribution are in good agreement with experimental results. Further statistical inference indicates that the E-P interactions prefer to modulate the mRNA level by controlling promoter activation and transcription initiation rates. Our model and results provide quantitative insights into both spatiotemporal gene-expression determinants (i.e., long-range E-P interactions) and cellular fates during development.

## Author summary

Gene expression is an essential biological process in all organisms. Numerous experimental studies have reported that the long-range enhancer-promoter (E-P) interaction on three-dimensional (3D) chromatin architecture plays important roles in regulating gene expression and cell functions, but the quantitative and qualitative impact of E-P interaction on gene expression over time is unclear. We develop a theoretically and numerically efficient model (called the 4D nucleome equation) to couple E-P interaction with gene expression and use this equation to characterize dynamic behavior. Then, we obtain the

GitHub (https://github.com/cellfate/
4DNucleomeEquation).

**Funding:** This work was supported by the National
Key R&D Program of China (2021YFA1302500 to
JZ), the Natural Science Foundation of P. R. China
(12171494 to JZ;11931019 to TZ; 11775314 to TZ;
62373384 to TZ; 12301646 to ZZ), the Guangdong
Basic and Applied Basic Research Foundation
(2022A1515011540 to JZ; 2023A1515011982 to
ZZ), Key-Area Research and Development Program
of Guangzhou, P. R. China (2019B110233002 to
JZ; 202007030004 to TZ), the Guangdong
Province Key Laboratory of Computational Science
at the Sun Yat-sen University (2020B1212060032
to JZ), the Fundamental Research Funds for the
Central Universities, Sun Yat-sen University
(23qnpy48 to ZW; 23qnpy49 to ZZ), and the China
Postdoctoral Science Foundation (2023M734061
to ZW). The funders had no role in study design,
data collection and analysis, decision to publish, or
preparation of the manuscript.

**Competing interests:** The authors have declared
that no competing interests exist.

theoretical distribution of mRNAs and predict the gene expression profiles under E-P regulations. Interestingly, we find that E-P interactions can induce bimodal and trimodal shapes of mRNA distribution. When applying this framework to mouse embryonic stem cell data to investigate the dynamical behaviors of E-P interaction and gene expression, we reproduce the experimentally measured E-P contact frequencies and mRNA distributions under different E-P interactions. Our results support the picture of an essential mechanism for explaining phenotypic diversity and cellular decision-making.

## Introduction

Gene expression is tightly related to three-dimensional (3D) genome conformation that may change over 1D time [1–4]. Specific DNA sequences–promoters and enhancers–orchestrate transcription in a highly complex and multilayered manner to ensure accurate spatiotemporal gene expression [1–4]. Many experimental studies have shown the importance of the roles of distal enhancers in ensuring reliable cell functioning and cellular decision-making [5–10]. However, the mechanism of how 3D chromatin organization (in particular 3D enhancer-promoter (E-P) interactions) in time (4D) shapes gene-expression dynamics still remains elusive.

Hierarchic genomic structures support various possible E-P topologies and the connection of upstream stochastic E-P interaction to downstream gene transcription [11–16]. Many efforts have been made to explore the essential factors of E-P interaction affecting transcription. Recent live-imaging measurements have provided clear evidence that E-P genomic distance effectively controls gene activities [6,10,17]. And a collection of experimental evidence has established that the E-P interaction strength significantly impacts gene expression levels [6,7,18,19]. For example, the *sna* shadow enhancer, whose strength is determined by chromatin inheritance, generates more bursts than the *sna* primary enhancer in *Drosophila* embryos [6], and the hormone or heavy metal exposure, which externally alters the E-P interaction strength [20,21], boosts the mRNA level of c-Fos gene with the increase of dose concentrations [18]. These experimental observations indicate that the E-P genomic distance and the E-P interaction strength are important factors impacting gene expression profiles. However, biological experiments alone are not sufficient to unravel the complete picture of the dynamics of E-P interaction-regulated gene expression, and it is necessary to develop biologically reasonable mathematical models to investigate the underlying mechanism.

The conventional modeling of gene expression kinetics is based on simple models such as the two-state model and multistate model [22–30], in which an implicit hypothesis is that chromatin behavior is frozen. Recent work adds the effects of chromatin structure to the model [10,31], but still ignores the spatiotemporal dynamic regulation of chromatin topologies [31–35]. So far, we still lack a mechanistic mathematical model that couples stochastic chromatin organizations and stochastic gene expression processes. Addressing this issue faces two challenges. First, genomic structures are stochastic at almost every level of organization, and this stochasticity is suggestively linked to gene transcription and finally affects transcriptional outcomes [14,32]. Overall, the temporal disconnection, as well as the stochasticity of E-P topology and gene expression, lies at the heart of a broad challenge in the physical biology of both establishing a comprehensive theoretical framework of the information transmission from the upstream chromatin organization to the downstream gene expression and forecasting mRNA profiles from the dynamics of underlying molecular processes. Second, the regulation of gene expression by the chromatin spatial structure is a multiscale system. Many experimental studies have indicated distinct timescale differences between upstream chromatin dynamics and

downstream gene expression [36,37]. For instance, E-P interaction occurs on a timescale of seconds to minutes [8,38,39], whereas the gene produces over a longer timescale compatible with E-P interaction, precisely the timescale of minutes to hours as suggested in multiple studies [9,36,40].

Here, we developed a general theoretical framework (formulated as the so-called 4D nucleome equation) to investigate how E-P interaction (characterized by E-P genomic distance and E-P interaction strength) affects gene expression dynamics. Specifically, this framework considered upstream chromatin motion on a fast timescale and downstream mRNA production on a slow timescale as well as the temporal connection between the upstream and the downstream. This framework involving space and time is referred as 4D gene expression kinetics [41,42]. With the timescale separation method, we derived analytical mRNA distribution and studied how E-P interaction qualitatively impacts the characteristics of mRNA distribution. Importantly, we found that the E-P interaction can flexibly regulate mRNA patterns by inducing multiple shapes of mRNA distributions including bimodal distribution with two non-origin peaks and even trimodal distribution. And our theoretical analysis and simulations revealed a power-law scaling of gene expression levels in the E-P genomic distance. Finally, by performing statistic inference on the mouse embryonic stem cells (mESCs) data, our framework exhibited good predictions of mRNA distribution and E-P contact probability under different E-P interactions. We emphasize that our 4D nucleome equation provides a general modeling framework for studying how chromatin dynamics affect gene expression kinetics and our results suggest a possible mechanism for explaining phenotypic diversity and cellular decision-making in realistic cases.

## Materials and methods

In this section, we introduce a theoretical framework for predicting gene expression regulated by long-range E-P interaction in 4D (Fig 1A). First, we model chromatin as a polymer [43–46] discretized into a collection of successive monomers, and assume that there are $N$ monomers on chromatin, spatial positions of which are denoted by $r = [r_1, \cdots, r_N]^T$. Each monomer represents a segment of DNA, whose length depends on the levels of coarse-grained chromatin. Second, we model the gene expression process as a discrete multistate model, and assume that the gene has $K$ different gene states (each state includes promoter's state (ON or OFF) and the number of mRNA), which altogether constitute the vector $s = [s_1, \cdots, s_K]^T$.

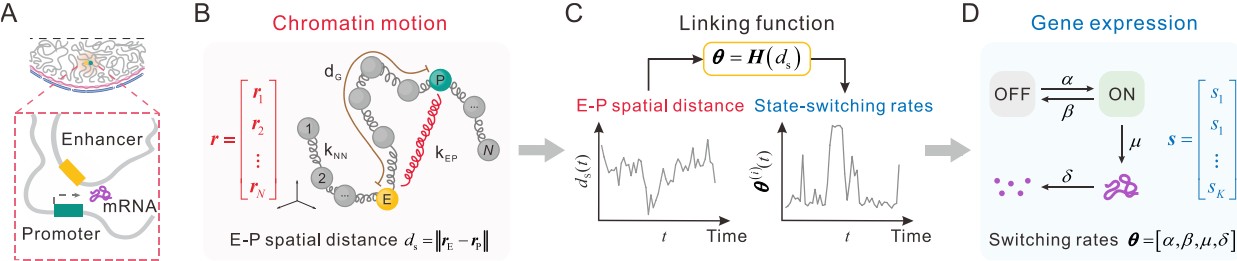

**Fig 1. Schematic representation of gene expression regulated by E-P interaction.** (**A**) E-P interaction in the cell nucleus plays a key role in regulating gene expression. (**B**) A polymer model involving E-P interaction (the red spring with coefficient $k_{EP}$) is proposed to simulate chromatin dynamics, where $r = [r_1, \cdots, r_N]^T$ represents monomers positions in 3D space, $d_G$ is E-P genomic distance, and $d_S = \|r_E - r_P\|$ is E-P spatial distance. (**C**) A link function vector $\theta = H(d_s)$, where $H$ is a function vector, bridges the temporal disconnection between upstream E-P topology and downstream gene expression. (**D**) The two-state telegraph model is used to imitate gene expression, where $s = [s_1, \cdots, s_K]^T$ is the vector of the gene's states and $\theta = [\alpha, \beta, \mu, \delta]$ is the vector of state-switching rates.

## 4D nucleome equation governing probabilistic evolutionary behavior in phase space

Let $\boldsymbol{p}(\boldsymbol{r},\boldsymbol{s};t)$ be a vector of the joint probability density functions that monomers are in position $\boldsymbol{r}$ and the gene is in state $\boldsymbol{s}$ at time $t$. Specifically, $\boldsymbol{p}(\boldsymbol{r},\boldsymbol{s};t) = [p(\boldsymbol{r},s_1;t),\cdots,p(\boldsymbol{r},s_K;t)]^{\mathrm{T}}$. Noting $\boldsymbol{p}(\boldsymbol{r},\boldsymbol{s};t) = p(\boldsymbol{r};t)\boldsymbol{p}(\boldsymbol{s}|\boldsymbol{r};t)$, we have

$$\frac{\partial \boldsymbol{p}(\boldsymbol{r},\boldsymbol{s};t)}{\partial t} = \frac{\partial p(\boldsymbol{r};t)}{\partial t}\boldsymbol{p}(\boldsymbol{s}|\boldsymbol{r};t) + \frac{\partial \boldsymbol{p}(\boldsymbol{s}|\boldsymbol{r};t)}{\partial t}p(\boldsymbol{r};t). \tag{1}$$

On one hand, the motion of monomers has a continuous trajectory in the region $\Omega$ (a connected and bounded domain). The derivative $\partial p(\boldsymbol{r};t)/\partial t$ in Eq [1] can be formally written as $\partial p(\boldsymbol{r};t)/\partial t = -\nabla_r \cdot [\boldsymbol{F}(\boldsymbol{r},\boldsymbol{s};t)p(\boldsymbol{r};t)]$, where $\nabla_r$ is the gradient operator and $\boldsymbol{F}(\boldsymbol{r},\boldsymbol{s};t)$ is a velocity field. Next, if we consider isotropic diffusion and friction across the region of monomers' positions, then $\boldsymbol{F}(\boldsymbol{r},\boldsymbol{s};t)$ takes a generalized Fokker–Planck approximation, i.e., $\boldsymbol{F}(\boldsymbol{r},\boldsymbol{s};t) = \boldsymbol{V}(\boldsymbol{r},\boldsymbol{s};t) -\nabla_r(D\log p(\boldsymbol{r};t))$, where the first term represents the deterministic part of the velocity field and the second term is a stochastic ingredient of the velocity under isotropic diffusion with diffusion coefficient $D$. We further assume that changes in gene state do not contribute to chromatin motion, and $\boldsymbol{V}(\boldsymbol{r},\boldsymbol{s};t)$ can be approximated by $\boldsymbol{V}(\boldsymbol{r};t)$. Thus, we can obtain

$$\frac{\partial p(\boldsymbol{r};t)}{\partial t} = -\nabla_r \cdot (\boldsymbol{V}(\boldsymbol{r};t)p(\boldsymbol{r};t)) + \nabla_r^2(Dp(\boldsymbol{r};t)), \tag{2}$$

where $\nabla_r^2$ is the Laplace operator.

On the other hand, the stochastic gene expression process regulated by chromatin dynamics can be modeled by the master equation of the form

$$\frac{\partial \boldsymbol{p}(\boldsymbol{s}|\boldsymbol{r};t)}{\partial t} = \boldsymbol{W}(\boldsymbol{r};t)\boldsymbol{p}(\boldsymbol{s}|\boldsymbol{r};t), \tag{3}$$

where $\boldsymbol{W}(\boldsymbol{r};t)$ is a monomer position-dependent state transition matrix, and the elements of $\boldsymbol{W}$ are related to gene state switching rates and chromatin position.

We assume that tiny changes in monomers coordinates do not alter gene state, implying that the derivative of conditional probability, $\nabla_r \boldsymbol{p}(\boldsymbol{s}|\boldsymbol{r};t)$, approximately equals zero partly because the time interval of gene state transition is generally longer than that of chromatin motion [36]. Thus, substituting Eq [2] and Eq [3] into Eq [1] yields the following equation

$$\frac{\partial \boldsymbol{p}(\boldsymbol{r},\boldsymbol{s};t)}{\partial t} = \underbrace{-\nabla_r \cdot (\boldsymbol{p}(\boldsymbol{r},\boldsymbol{s};t)\boldsymbol{V}(\boldsymbol{r};t)^{\mathrm{T}}) + \nabla_r^2(D\boldsymbol{p}(\boldsymbol{r},\boldsymbol{s};t))}_{\text{chromatin dynamics}} + \underbrace{\boldsymbol{W}(\boldsymbol{r};t)\boldsymbol{p}(\boldsymbol{r},\boldsymbol{s};t)}_{\text{gene−expression dynamics.}} \tag{4}$$

The first two terms on the right-hand side of Eq [4] represent chromatin's spatiotemporal diffusion process. The first term is the deterministic component, and the second is the stochastic component accounting for random fluctuations. The last term captures the gene states' random switching process. It should be noted that Eq [4] is a comprehensive description of the gene expression process toward the 4D reality, so we called it the 4D nucleome equation. In the Results section, we will derive the mRNA distribution based on Eq [4].

## Stochastic model simulating trajectory evolutionary behavior in configuration space

**Modeling chromatin dynamics.** Successive monomers in the chromatin are connected with harmonic springs of stiffness $k_{\mathrm{NN}}$ (Fig 1B). Each monomer represents a nucleosome or a

DNA segment (this length does not affect our qualitative results to be obtained) with the 3D position denoted by $\boldsymbol{r}_i = (r_{i1}, r_{i2}, r_{i3})$, where $i = 1,\ldots,N$. We employ one nucleosome to represent an enhancer or promoter and posit that there are only one enhancer and one promoter on the chromatin. To mimic E-P interaction and to simplify without loss of generality, we add a harmonic spring with stiffness $k_{EP}$ between the enhancer and the promoter to materialize this abstract concept.

Based on the above assumptions, we model the chromatin dynamics according to the Langevin equation $d\boldsymbol{r} = \boldsymbol{V}(\boldsymbol{r};t)dt + \sqrt{2D}d\boldsymbol{B}(t)$, which is equivalent to Eq [2], where $B(t)$ is a vector of independent Brownian motions. And $\boldsymbol{V}(\boldsymbol{r};t) = -\nabla_r U(\boldsymbol{r};t)/\gamma$, where $\gamma$ is the friction coefficient and $U(\boldsymbol{r};t)$ is the total potential of chromatin conformation. Note that

$U(\boldsymbol{r};t) = U_{NN}(\boldsymbol{r};t) + U_{EP}(\boldsymbol{r};t)$, where without loss of generality, we set $U_{NN}(\boldsymbol{r};t) =$

$(1/2)\sum_{j=1}^{N-1} k_{NN}(\boldsymbol{r}_j - \boldsymbol{r}_{j+1})^2$ which represents the potential for the chain connection, and

$U_{EP}(\boldsymbol{r};t) = (1/2)k_{EP}(\boldsymbol{r}_E - \boldsymbol{r}_P)^2$ which represents the potential for the E-P interaction with E, P$\in\{1,\cdots,N\}$ representing the index of the monomer occupied by enhancer and promoter respectively. We let $d_G = |E-P|$ represent the E-P genomic distance, which can be directly measured by an experimental method. However, the $k_{EP}$ representing the E-P interaction strength cannot be directly measured by experiments but can be estimated from experimental data, e.g., Hi-C data [47] or Capture-C data [10].

**Modeling gene expression dynamics.** Gene expression process can be characterized by a two-state model with an active ON and a silent OFF state (Fig 1D). The switching rates from OFF to ON and vice versa are $\alpha$ and $\beta$, respectively. The transcription rate is $\mu$, and the mRNA degradation rate is $\delta$. We define $\boldsymbol{\theta} = [\alpha,\beta,\mu,\delta]$ as the gene-state switching rates set.

**Bridging chromatin dynamics and gene expression.** After having identified chromatin conformations and transcriptional reactions independently, we next build a biologically reasonable link between them (Fig 1C). E-P interaction carries regulatory information to orchestrate gene expression, and we hypothesize that the time-varying E-P spatial distance $d_S = \|\boldsymbol{r}_E - \boldsymbol{r}_P\|$ encodes the information to regulate the transcription-associated reaction rates $\boldsymbol{\theta}$. That is $\boldsymbol{\theta} = \boldsymbol{H}(d_s)$, where $\boldsymbol{H}$ is a link function vector. Therefore, the two-state model mentioned above becomes a variable two-state model in which the rates $\boldsymbol{\theta}$ depend on the E-P spatial distance $d_S$.

Recent experimental results show that E-P proximity increases the likelihood of gene expression [14]. It is reasonably assumed that if $d_S$ is smaller, then $\alpha$ and $\mu$ are larger, but $\beta$ is smaller (in the case that $\delta$ is constant). For simplification, we assume $\beta$ is independent of $d_S$ and $\alpha(\mu)$ changes between $\alpha_{max}(\mu_{max})$ and $\alpha_{min}(\mu_{min})$ with the fluctuating $d_S$. We choose a Hill function [48] to reflect the effect of changes in the E-P spatial distance $d_S$ on the rates $\boldsymbol{\theta}$, although the specific shape of the corresponding response curve can also be captured by other functions. Specifically, we assume that the rates $\alpha$ and $\mu$ depend nonlinearly on $d_S$, i.e., each is a Hill-like function vector $\boldsymbol{H}$ (see S1 Text).

Under the above settings, we propose a stochastic simulation algorithm to simulate the time evolution of the entire system (see S1 Text). In Result section, we will study how E-P interaction strength $k_{EP}$ and E-P genome distance $d_G$ regulate gene expression and how they affect the shape and characterization of mRNA distributions. In short, our modeling strategy provides a possible framework for characterizing 3D chromatin motion and tracking gene-expression processes over 1D time [41,42].

## Model-data approach to infer E-P interactions and gene-expression kinetics

We use the above model to fit the mRNA distribution of single-molecule RNA fluorescence in situ hybridization (smRNA–FISH) in mESCs [10]. The data includes 6 cell lines $C_k$

($k = 1,\ldots,6$). In each cell line, the enhancer is placed at different positions from the promoter to drive the expression of enhanced green fluorescent protein (eGFP), so we can get the E-P genomic distance and corresponding mRNA distribution with the bin number $n_k$ as well as the corresponding steady-state probability distribution $Q_k(X = x_i)$ ($1 \leq i \leq n_k$).

For each cell line $C_k$, we theoretically calculate the steady-state probability distribution with parameters $\mathbf{\Gamma}$ (including E-P interaction parameters and gene expression parameters). Then, we discretize the steady-state distribution $P_k(X = x_i; \mathbf{\Gamma})$ ($1 \leq i \leq n_k$) that is comparable with the experiment data $Q_k(X = x_i)$. Note that the cross entropy of the cell line $C_k$ is given by

$$H_k(\mathbf{\Gamma}) = -\sum_{i=1}^{n_k} Q_k(X = x_i)\log(P_k(X = x_i; \mathbf{\Gamma})), \tag{5}$$

and the best-fit parameters by minimizing the total cross entropy function

$$\arg\min_{\mathbf{\Gamma}} H(\mathbf{\Gamma}) = \arg\min_{\mathbf{\Gamma}} -\sum_{k=1}^{6}\sum_{i=1}^{n_k} Q_k(X = x_i)\log(P_k(X = x_i; \mathbf{\Gamma})). \tag{6}$$

In fact, the minimum cross-entropy method is the same as maximum likelihood estimation. To solve this optimization problem, we use the *fmincon* function (a nonlinear programming solver) in the LBFGS method of MATLAB to find the minimum value of the optimization problem given a set of initial values and parameter intervals (S1 Text).

## Results

### Analytical results of steady mRNA distribution

To study the qualitative and quantitative effects of E-P interaction (including E-P genomic distance $d_G$ and E-P interaction strength $k_{EP}$) on gene expression (including the mRNA distribution, the mean mRNA expression level and the coefficient of variation (CV) defined as the ratio of standard deviation over mean), we use the 4D nucleome equation (Eq [4]) to solve the steady mRNA distribution $P(x)$ depending on the E-P spatial distance $d_S$.

Note that the entire gene expression system contains two modules–upstream chromatin conformation on a fast timescale and downstream gene expression on a slow timescale. If the typical timescales for the former and latter are denoted by $\tau_{up}$ and $\tau_{down}$ respectively, Eq [4] can be rewritten as the rescaled equation

$$\frac{\partial \boldsymbol{p}(\boldsymbol{r}, \boldsymbol{s}; t)}{\partial t} = -\frac{1}{\tau_{up}}\left[\nabla_{\boldsymbol{r}} \cdot (\boldsymbol{p}(\boldsymbol{r}, \boldsymbol{s}; t)\tilde{\boldsymbol{V}}(\boldsymbol{r}; t)^{\mathrm{T}}) + \nabla_{\boldsymbol{r}}^2(\tilde{\boldsymbol{D}}\boldsymbol{p}(\boldsymbol{r}, \boldsymbol{s}; t))\right] + \frac{1}{\tau_{down}}\left[\tilde{\boldsymbol{W}}(\boldsymbol{r}; t)\boldsymbol{p}(\boldsymbol{r}, \boldsymbol{s}; t)\right], \tag{7}$$

where $\tilde{\boldsymbol{V}} = \tau_{up}\boldsymbol{V}$, $\tilde{\boldsymbol{D}} = \tau_{up}\boldsymbol{D}$, $\tilde{\boldsymbol{W}} = \tau_{down}\tilde{\boldsymbol{W}}$. Deriving the mRNA distribution directly from the above multiscale system is a particular challenging task. Here we resort to a timescale separation method. Before this, however, we consider the E-P interaction and gene expression dynamics independently.

When focusing only on the motion of upstream chromatin conformation, we find that the stationary probability density function of chromatin conformation $\boldsymbol{r}$ can be expressed as $p^{\mathrm{stat}}(\boldsymbol{r}) = \prod_{i=1}^{3} p_i(\boldsymbol{r})$ since every monomer moves independently in each dimension. Notably, we find that E-P spatial distance $d_S$ obeys the following exact Maxwell-Boltzmann distribution (Fig 2A and S1 Text)

$$p_{DS}(d_S) = \sqrt{\frac{2}{\pi}}\Theta^{-3}d_S^2\exp\left(-\frac{d_S^2}{2\Theta^2}\right), \tag{8}$$

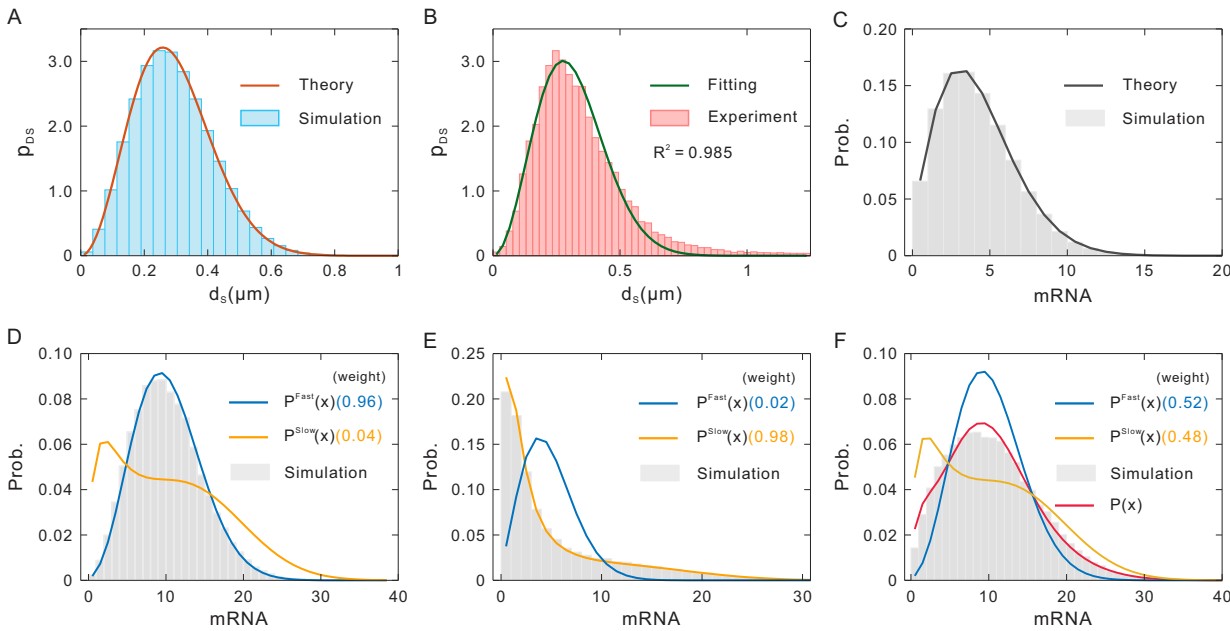

**Fig 2. Distributions of gene expression model.** (**A**) The distribution of E-P spatial distance (Maxwell-Boltzmann distribution given by Eq [8]) where red lines represent theoretical results, and histograms represent numerical results. (**B**) Experimental measurement of 3D spatial distance between Sox2 and its enhancer in living ESCs (histogram, [49]). The blue line is obtained by fitting Eq [8]. (**C**) The mRNA distribution conditional on frozen chromatin structure (Poisson-Beta distribution) where $\alpha = 0.30$, $\beta = 0.50$, $\mu = 1$ and $\delta = 0.1$. (**D**) Upstream chromatin fluctuations are much faster compared with the downstream state switching rates ($\omega = 0.04$), where the values in parentheses indicate the weight of the interpolation distribution. Parameter values are set as $\gamma = 0.1$, $k_{EP} = 0.3$. (**E**) Upstream chromatin fluctuations are slow ($\omega = 59.95$). Parameter values are set as $\gamma = 100$, $k_{EP} = 0.1$. (**F**) Upstream chromatin fluctuations and downstream state switching rates are comparable ($\omega = 0.91$). Parameter values are set as $\gamma = 5$, $k_{EP} = 0.3$. Other parameters in (D-F) are set as $\alpha_{max} = 0.60$, $\alpha_{min} = 0.30$, $\beta = 0.50$, $\mu_{max} = 3$, $\mu_{min} = 0.1$, $\delta = 0.1$.

where the lumping parameter $\Theta = \sqrt{D\gamma(k_{NN}/d_G + k_{EP})^{-1}}$ determines how changes in E-P interaction strength $k_{EP}$ or E-P genomic distance $d_G$ alter the shape of the $p_{DS}(d_S)$. Experimental measurement of 3D spatial distance between Sox2 and its enhancer in living ESCs [49] or between the *even-skipped* (*eve*) locus and its enhancer in the fly embryo [50] can be well fitted by Eq [8], confirming the validity of theoretical analysis (Fig 2B).

When focusing only on the downstream gene expression, that is, given a $d_S$ (or a frozen E-P topology), we find that the stationary probability distribution of mRNA abundance $P_{mRNA}(x)$ is a Poisson-Beta distribution (Fig 2C, [51]). That is,

$$P_{mRNA|d_S}(x) = \text{Poisson}(x; \tilde{\mu}t) \underset{t}{\wedge} \text{Beta}(t; \tilde{\alpha}, \tilde{\beta})$$

$$= \int_0^1 \frac{(\tilde{\mu}t)^x}{x!} e^{-\tilde{\mu}t} \frac{t^{\tilde{\alpha}-1}(1-t)^{\tilde{\beta}-1}}{B(\tilde{\alpha}, \tilde{\beta})} dt, \tag{9}$$

where $\wedge$ represents the mixture of two distributions, and the gene-state switching rates are in the unit of the mRNA decay rate, i.e., $\tilde{\alpha} = \alpha/\delta$, $\tilde{\beta} = \beta/\delta$, $\tilde{\mu} = \mu/\delta$ (Fig 2C).

Now, we turn to derive mRNA distribution by using the timescale separation method. First, we assume that the upstream process is much faster than the downstream process, i.e., we only consider the limit case of $\tau_{up}/\tau_{down} = \varsigma \ll 1$. By rescaling $t$ to $t/\tau_{down}$, we can then assume the formal solution $p(\mathbf{r}, \mathbf{s}; t) = \mathbf{p}^{(0)}(\mathbf{r}, \mathbf{s}; t) + \varsigma \mathbf{p}^{(1)}(\mathbf{r}, \mathbf{s}; t) + O(\varsigma^2)$. Considering the leading order of $\varsigma$ and the stationary condition, we have $0 = \nabla_{\mathbf{r}} \cdot (\mathbf{p}^{(0)}(\mathbf{r}, \mathbf{s}; t) \tilde{\mathbf{V}}(\mathbf{r}; t)^T) + \nabla_{\mathbf{r}}^2 (\tilde{\mathbf{D}} \mathbf{p}^{(0)}(\mathbf{r}, \mathbf{s}; t))$.

Since we have neglected the regulation of gene state by chromatin structure, the zero-th order solution $\boldsymbol{p}^{(0)}(\boldsymbol{r},\boldsymbol{s};t)$ can be decoupled as $\boldsymbol{p}^{(0)}(\boldsymbol{r},\boldsymbol{s};t) = p^{\text{stat}}(\boldsymbol{r})\boldsymbol{p}(\boldsymbol{s};t)$, implying that the quasi-stationary of chromatin motion is well separated from the downstream transcription. In general, the stationary mRNA probability distribution is given through (Fig 2D)

$$P^{\text{Fast}}(x) = P_{\text{mRNA}|\langle DS\rangle}(x|\int_{0}^{+\infty} \boldsymbol{H}(d_{\text{S}})p_{DS}(d_{\text{S}})dd_{\text{S}})$$
$$= \text{Poisson}(x; \langle\tilde{\mu}(d_{\text{S}})\rangle t)\underset{t}{\wedge}\text{Beta}(t; \langle\tilde{\alpha}(d_{\text{S}})\rangle, \langle\tilde{\beta}(d_{\text{S}})\rangle),$$
(10)

where $\langle DS\rangle$ is the expectation of E-P spatial distance and $P_{\text{mRNA}|\langle DS\rangle}(x)$ (the distribution conditional on the averaged $d_{\text{S}}$) is given through Eq [9].

Second, we consider the opposite limit, i.e., $\tau_{up}/\tau_{down} = \zeta \gg 1$. In this limit and similar to the analysis in the case of $\tau_{up}/\tau_{down} = \varsigma \ll 1$, we find that $\boldsymbol{p}^{(0)}(\boldsymbol{r},\boldsymbol{s};t)$ can also be decoupled as $\boldsymbol{p}^{(0)}(\boldsymbol{r},\boldsymbol{s};t) = p(\boldsymbol{r};t)\boldsymbol{p}^{\text{stat}}(\boldsymbol{s}|\boldsymbol{r})$, where the equation group $\tilde{\boldsymbol{W}}(\boldsymbol{r})\boldsymbol{p}^{\text{stat}}(\boldsymbol{s}|\boldsymbol{r}) = \boldsymbol{0}$ determines $\boldsymbol{p}^{\text{stat}}(\boldsymbol{s}|\boldsymbol{r})$. Therefore, the chromatin spatial positions are not independent of the downstream gene expression. But the stationary mRNA probability distribution can be calculated according to (Fig 2E)

$$P^{\text{Slow}}(x) = \int_{0}^{+\infty} P_{\text{mRNA}|d_{\text{S}}}(x|\boldsymbol{H}(d_{\text{S}}))p_{DS}(d_{\text{S}})dd_{\text{S}}$$
$$= \text{Poisson}(x; \tilde{\mu}(d_{\text{S}})t) \underset{t}{\wedge} \text{Beta}(t; \tilde{\alpha}(d_{\text{S}}), \tilde{\beta}(d_{\text{S}})) \underset{d_{S}}{\wedge} \text{MaxBoltz}(d_{\text{S}})$$
(11)

where $P_{\text{mRNA}|d_{\text{S}}}(x)$ (i.e., the distributions conditional on the $d_{\text{S}}$) is given by Eq [9].

Finally, in the intermediate regime, we find that $P(x)$ can be well fit with the following formula (Fig 2F)

$$P(x) \approx \frac{1}{1+\omega}P^{\text{Fast}}(x) + \frac{\omega}{1+\omega}P^{\text{Slow}}(x)$$
$$= \frac{1}{1+\omega}\text{Poisson}(x; \langle\tilde{\mu}(d_{\text{S}})\rangle t) \underset{t}{\wedge} \text{Beta}\left(t; \langle\tilde{\alpha}(d_{\text{S}})\rangle, \langle\tilde{\beta}(d_{\text{S}})\rangle\right)$$
$$+ \frac{\omega}{1+\omega}\text{Poisson}(x; \tilde{\mu}(d_{\text{S}})t) \underset{t}{\wedge} \text{Beta}\left(t; \tilde{\alpha}(d_{\text{S}}), \tilde{\beta}(d_{\text{S}})\right) \underset{d_{\text{S}}}{\wedge} \text{MaxBoltz}(d_{\text{S}}),$$
(12)

where $\omega$ is a scaling factor defined as the ratio of minimum variable transition rate and maximum chromatin motion velocity. $\omega = \min\{\alpha_{\min}, \mu_{\min}\}/\max V(\boldsymbol{r})$, where $V(\boldsymbol{r}) = [(k_{\text{NN}}/d_{\text{G}} + k_{\text{EP}})d_{\text{S}}]/(b\gamma)$, $b$ is E-P encounter distance and $d_{\text{S}}$ can be selected as the maximum E-P spatial distance that can be theoretically reached (in fact, the cumulative density function of $p_{DS}(d_{\text{S}})$ reaches 0.99).

It is worth mentioning that Eq [12] provides high-accuracy approximations of mRNA steady distribution and is a useful and simple formula for predicting the dynamics of mRNA over a broad range of timescale separation. We can further trace the respective contributions of the system's key parameters (e.g., $d_{\text{G}}$ and $k_{\text{EP}}$) to the patterns of mRNA distribution, to better reveal the essential mechanism of gene expression.

## E-P interaction controls the mean level and variability of gene expression

To understand how E-P interaction modulates gene expression, we use the above theoretical analysis to explore the qualitative impact of E-P genomic distance $d_{\text{G}}$ and E-P interaction strength $k_{\text{EP}}$ on gene expression. We calculate the mean ($\langle\text{mRNA}\rangle$) and the CV of the mRNA probability distribution in steady state, based on Eq [12].

Fig 3A and 3B depict how changes in $d_{\text{G}}$ alter $\langle\text{mRNA}\rangle$ and CV. We find that when $k_{\text{EP}}$ is fixed, $\langle\text{mRNA}\rangle$ monotonously decreases and CV monotonically increases with increasing $d_{\text{G}}$.

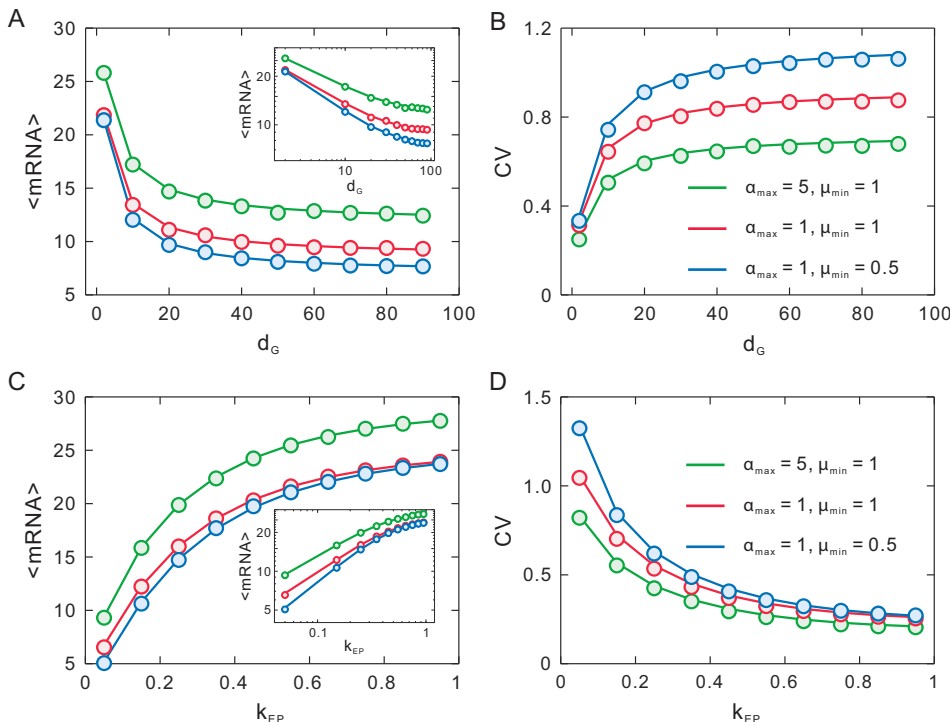

**Fig 3. Influence of E-P interaction on mRNA mean and CV. (A-B)** The effect of E-P genomic distance $d_G$ on mRNA mean and CV, where solid lines represent theoretical results, and color points represent numerical results. **(C-D)** The effect of E-P interaction strength $k_{EP}$ on mRNA mean and CV. The inset in (A) and (C) shows the log-log plot. Other parameter values are set as $\alpha_{min} = 0.06$, $\beta = 0.20$, $\mu_{max} = 3$, $\delta = 0.1$, $\gamma = 50$.

Theoretically, a smaller $d_G$ corresponds to a shorter E-P spatial distance and a more frequent E-P interaction, thus boosting the expression level. Fig 3C and 3D reveal the distinct effects of different $k_{EP}$ on gene expression. It shows that the tendencies are completely different from those in the case of changing $d_G$. ⟨mRNA⟩ is monotonously upregulated and CV is downregulated by $k_{EP}$. Indeed, Eq [8] has shown the opposite effect of $d_G$ and $k_{EP}$ on E-P spatial distance and then on gene expression. Log-log plot represents power-law behaviors of ⟨mRNA⟩ regulated by $d_G$ or $k_{EP}$ (Fig 3A and 3C inset). Moreover, it can be seen that the enhancement or the reduction of ⟨mRNA⟩ (at larger $d_G$ or $k_{EP}$) tends to be saturated, indicating that the effect of E-P interaction on gene expression is not infinite but limited.

Notably, the qualitative behaviors of ⟨mRNA⟩ are consistent with many experimental observations [6,10,17,18,52]. For example, a larger $d_G$ between *sna* shadow enhancer and promoter generates fewer mRNAs than a smaller $d_G$ [6,17], and a stronger enhancer (corresponding to a larger $k_{EP}$) produces more mRNAs than a weaker enhancer [6] in living *Drosophila* embryos. And with the increase of dose concentrations (corresponding to the increment of $k_{EP}$), the mRNA of the MS2 reporter gene grows and reaches saturation [52]. Another example is that in mESCs, the mRNA level decreases and tends to remain unchanged in response to the enlargement of E-P genomic distance [10].

In addition, we study the combined effect of state-switching rates and E-P interaction on gene expression. The downstream gene-expression model is a variable ON-OFF model with variable $\alpha$ and $\mu$. We find that increasing $\alpha$ (or $\mu$, no matter increase $\alpha_{max}$ (or $\mu_{max}$) and $\alpha_{min}$ (or $\mu_{min}$)) enlarges ⟨mRNA⟩ and reduces CV (Fig 3). This is because larger $\alpha$ can elongate the

ON state residence time, and larger $\mu$ makes the time interval of mRNA generation become shorter, and each case leads to enlargement in $\langle$mRNA$\rangle$ and diminishment in CV.

## E-P interaction can induce bimodal and multimodal mRNA distributions

Having clarified the qualitative effect of E-P interaction on the $\langle$mRNA$\rangle$ and CV of mRNA, we next analyze how E-P interaction impacts the shape of mRNA distribution, including the peak numbers and peak probabilities. We adjust the E-P interaction strength $k_{EP}$ and E-P genomic distance $d_G$ and calculate the theoretical distributions according to Eq [12].

Each peak is defined as the observed local maximum value to identify the peak number of mRNA distributions. Based on this, we draw the boundary lines. As a result, we find that the unimodal (U), bimodal (B), and trimodal (T) distributions can appear (Fig 4A). Besides, the heatmap in Fig 4A is the bimodal coefficient $BC = 1/(K-S^2)$, where $S = v_3/v_2^{3/2}$ is the skewness of mRNA distribution, $K = v_4/v_2^2$ is the kurtosis, and $v_i$ is the $i$-th central moment, $i = 2,3,4$.

Fig 4B shows that when $d_G$ is fixed, with the increase of $k_{EP}$, the distribution can appear in five modes: unimodal with the origin peak (OP), bimodal with one OP and one non-origin peak (NOP), trimodal with one OP and two NOPs, bimodal with two NOPs and unimodal with the NOP. In fact, we can see the evolutionary process of the peak numbers and peak probabilities. The OP goes under with amplifying the $k_{EP}$, and a NOP begins to grow and a bimodal then occurs. When the distance between the two peaks of the bimodal becomes larger, a peak emerges at the larger mRNA and the bimodal distribution turns to trimodal distribution. Subsequently, the OP vanishes and gets back to the bimodal. The distance between the two NOPs of the bimodal distribution can become smaller, and the peak corresponding to the fewer mRNA disappears when the bimodal becomes unimodal. Fig 4D perfectly demonstrates the above process, and the simulations (histograms) are in good agreement with theoretical predictions (solid lines). In addition, when we fix $k_{EP}$, the influence of $d_G$ on the distribution

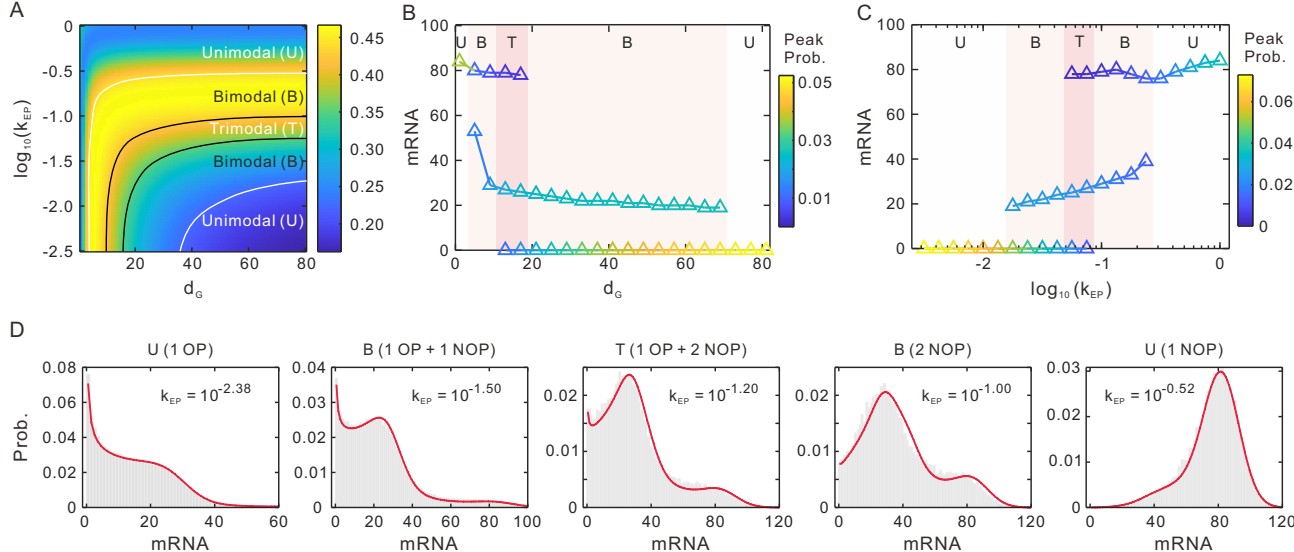

**Fig 4. E-P interaction can induce multiple shapes of mRNA distribution. (A)** Effects of E-P interaction on mRNA distribution. The lines stand for the boundaries of different peak numbers. The plain is divided into distinctive regions representing unimodal (U), bimodal (B) and trimodal (T), respectively. The heatmap is for the bimodal coefficient. **(B)** Dependence of the most probable mRNA numbers on $k_{EP}$, where $d_G = 70$. The color regions represent different peak numbers. The color bar represents peak probability. **(C)** Dependence of the most probable mRNA numbers on $d_G$, where $k_{EP} = 10^{-1.76}$. **(D)** The example of unimodal/bimodal/trimodal mRNA distribution in (B). OP (origin peak), NOP (non-origin peak). The solid lines represent theoretical results, and histograms represent numerical results. Parameter values are set as $d_G = 70$, $\alpha_{max} = 2$, $\alpha_{min} = 0.06$, $\beta = 0.10$, $\mu_{max} = 9$, $\mu_{min} = 3$, $\delta = 0.1$, $\gamma = 50$.

modes is opposite to that of $k_{EP}$ (Comparing Fig 4B and 4C). It should be pointed out that although the trimodal distribution may be absent under some parameter values, the pattern of distribution along $k_{EP}$ or $d_G$ still changes from unimodal to bimodal and then back to unimodal. During the bimidal phase, the peak at the smaller mRNA gradually disappears, resulting in the transition from bimodal to unimodal (see S2 and S3 Figs).

These results indicate that E-P interaction can produce multiple modes of mRNA distribution. More importantly, it can lead to mRNA distributions with two NOPs and even with three peaks. S1 Fig shows more bimodal and trimodal cases under different parameter values. Notably, only the downstream ON-OFF model (Eq [9]) can produce neither bimodal with two NOPs nor trimodal. In fact, the phenomenon of bimodal with two NOPs has been confirmed in biological experiments, such as the random activity of latent HIV-1 promoters [53]. In Ref [54],the researchers studied the transcriptional behavior of GREB1 changes with estrogen dose. Generally, addition of estrogen (17b-estradiol, E2) may increase contacts between the GREB1 gene and the estrogen-receptor-$\alpha$-bound enhancer [55], thus altering the E-P interaction strength. When increasing the E2 concentration (increasing the E-P interaction strength), the mRNA level grows and the mRNA distribution changes from the single peak to a bimodal distribution. A new peak emerges at a higher mRNA level and gradually increases and the peak near the origin gradually decreases, which is consistent with our model (see S2 Fig). In a word, these phenomena highlight the importance of E-P interaction in regulating gene expression, which may explain the source of multimodality. And the distinctive modes of mRNA distribution for achieving fast responses to stimulations and phenotypic switching would be essential for environmental adaptation and cellular decision-making.

## Analysis of mouse embryonic stem cell data indicates that E-P interaction regulates promoter activation and transcription initiation

To check the effectiveness of our model, we used different E-P distances that forming distinctive cell lines $C_k$ ($k = 1,\ldots,6$) and corresponding mRNA distribution measured by smRNA-FISH in mESCs [10] to infer gene expression dynamics (Materials And Methods). Assume that the mRNA distributions in different cell lines $C_k$ are different only due to the different E-P genomic distances, and that an enhancer is inserted into the upstream or downstream of an promoter with the same genomic distance has no significant difference. The E-P genomic distances for different cell lines are NaN, 112.1710 kb, 39.4530 kb, 23.1110 kb, 17.0190 kb and 6.0600 kb respectively, where the NaN means the eGFP transcription is driven by the Sox2 promoter alone. Then, we suppose that one monomer in the coarse-grained polymer model represents 5kb, the E-P genomic distances for our model in different cell lines $C_k$ are NaN, 22, 8, 5, 3, and 1, respectively. It should be pointed out that the $d_G$ is the value after rounding.

Frist, we preliminarily attempt to use the two-state model without considering E-P regulatory to infer gene expression dynamics. We find that the changing $\alpha$ and $\mu$ in different cell lines get the minimize cross entropy (Table A in S1 Text). Then, we involve the long-range E-P regulatory into the model and define three variable two-state models (the variable $\alpha$ two-state model, the variable $\mu$ two-state model, and the variable $\alpha$ and $\mu$ two-state model) to simulate the transcription process (S1 Text). For each cell line $C_k$, we calculate the steady-state probability distributions by using Eq [12] and use Eq [6] to estimate the optimized parameter values (Materials And Methods). We find that the two-state model with variable $\alpha$ and $\mu$ obtains the minmum cross entropy, which indicates E-P communication prefers to regulate the promoter activation ($\alpha$) and transcription initiation ($\mu$) rates (Table B in S1 Text). The multiscale model fit converged to a good set of parameters $\mathbf{\Gamma} = [k_{NN}, k_{EP}, \gamma, \alpha_{min}, \alpha_{max}, \beta, \mu_{min}, \mu_{max}]$ describing

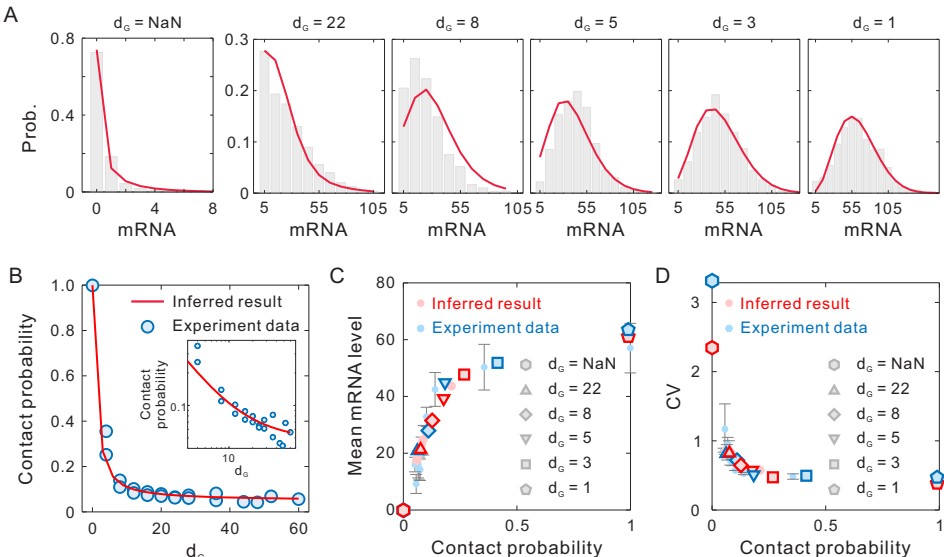

**Fig 5. Results obtained by analyzing experiment data on mESCs in response to the varying E-P genomic distance.**
**(A)** Distributions of the mRNA numbers of smRNA-FISH in different cell lines. The solid line shows the best fit of our stochastic multiscale model to the experimental data shown in the histograms. $d_G$ = NaN represents promoter-only control cell line. Parameter values are listed in Table E in S1 Text. **(B)** The relationship between $d_G$ and contact probability. The blue circles represent the experimental data from the ectopic Sox 2 transgene. The solid line is obtained based on the best-fit values of parameters $k_{NN}$ and $k_{EP}$. The inset shows the log-log plot. **(C)** Mean eGFP mRNA level plotted against contact probability between the ectopic Sox2 promoter and SCR insertions. The shadow blue dots show the experimental data presented as mean values +/- standard deviation, whereas the shadow red dots show the theoretical results obtained by best-fit parameter values. The blue polygon shows the experimental data in (A), and the red polygon shows the corresponding theoretical results. **(D)** CV of eGFP level against contact probability. The meanings of symbols are the same as those in (C).

chromatin and gene expression dynamics (see Table C in S1 Text). In addition, different monomer lengths do not alter the fitting results (see Table D in S1 Text).

As a result, Fig 5A shows that the 4D nucleome equation can well fit the mRNA distributions under different E-P interaction. It should be pointed out that the weight of slow chromatin dynamics ($1/(1+\omega)$) plays an important role in modifying the fitted distribution (see S5 Fig and S1 Text). Based on the best fitting parameter values and Eq [8], we can further study the relationship between E-P genomic distance and E-P contact probability (see S1 Text). Fig 5B shows that with increasing E-P genomic distance, the contact probability steeply decays and tends to be flat, which is in good agreement with the experimental results confirmed using Capture-C analysis [10]. Log-log plot represents a power-law behavior between E-P genomic distance and contact probability (Fig 5B inset), also in accordance with many experimental data in consecutive and nonconsecutive TAD borders [56], even on the genome-wide scale [57,58]. Second, Fig 5C shows the relationship between contact probability and mean gene expression level. We see that the values and the change tendency of the contact probability and average mRNA level obtained by theoretical calculations (light red circle) are basically consistent with the experimental data (light blue circle). If we focus on the six special cell lines shown in Fig 5A, we find that the inferred results (red polygon) and experimental data (blue polygon) are relatively consistent in values and change tendency. Third, for the relationship between CV of eGFP levels and contact probability (Fig 5D), we find that the theoretical result and experimental data are also relatively consistent in values and change tendency. In addition, we consider changing the upstream-downstream nonlinear Hill function in the model to a linear

form (i.e., using a linear function within a reasonable range of E-P spatial distances). By using the same fitting method, we find that the linear model fits the experimental data well (see S6 Fig). Therefore, in our model, non-linear dependence of the activation rate on E-P contacts does not seem to be a dominant cause of the nonlinear relationship between connection probability and mean expression observed in the experiment. Rather, the coupling of the fast-slow chromatin dynamics may be an important factor contributing to the nonlinearity.

Finally, we verify that the inferred results are consistent with previous observations. The inferred $k_{NN} = 0.7758$ is one order of magnitude larger than $k_{EP} = 0.0969$, indicating that the ectopic enhancer and promoter do not have a prominent specific interaction compared to the connection between successive monomers. This result is consistent with experimental studies [10]. Meanwhile, the fold change between $\alpha_{max}$ and $\alpha_{min}$ (18 folds) is larger than that between $\mu_{max}$ and $\mu_{min}$ (8 folds), indicating that E-P interaction is inclined to adjust OFF state dwell time (related to $\alpha$) and further verifying that E-P interaction mainly regulates burst frequency [6,7,10,59,60]. According to the inferred results, the ON-state dwell time is in the timescale of minutes, and the OFF-state dwell time is from minutes to hours [61–63] (considering that the time unit is mRNA lifetime, expected to be around 1.5 hours [64]). The number of mRNA per burst is about tens of mRNAs, consistent with a previous experimental observation [63].

Taken together, our model results are consistent with previous experimental observations and suggest a possible essential mechanism of gene expression regulated by E-P interaction.

## Discussion

Imaging studies, high-resolution chromatin conformation maps, and genome-wide occupancy profiles of architectural proteins have revealed that genome topology encoding E-P interaction information is tightly correlated with gene expression. In this paper, we have proposed a general yet tractable multiscale model, which integrates the E-P interaction information into gene expression, to shed light on the pivotal role of E-P spatial interaction in the control of gene expression profiles.

This theoretical framework characterizes the dynamic process of stochastic gene expression regulated by chromatin movement in the nucleus by using a 4D nucleome equation. First, the fast-scale chromatin motion processes (including E-P interaction) can be described by a generalized Rouse model, which is suitable for modeling the situation that the environmental effects of entanglement and crowding are negligible [65]. Second, the slow-scale gene expression process is described by a two-state model that captures essential events occurring in transcriptional processes. Third, an input-output relation was proposed to link upstream chromatin configurations to downstream gene expression. This equation unifies two stochastic processes with two different time scales, which can be taken as a good starting point for analyzing how chromatin motion affects gene expression in complex cases.

Overcoming time-scale differences and biological-process complexity to solve mRNA distribution analytically in this 4D nucleome equation is challenging. However, the analytical methodology developed here can be used in quantitatively studying distribution characteristics of gene expression involving E-P interaction. In fact, we have used the timescale separation method to obtain analytical mRNA distributions. In addition, our approach also allows us to make meaningful predictions about how upstream chromatin dynamics affect downstream gene-expression phenotypes. First, we have shown that E-P genomic distance $d_G$ and E-P interaction strength $k_{EP}$, two key parameters in our model, have the opposite change trends on mRNA mean levels and CVs, which were qualitatively verified by experimental data. Second, our model demonstrated that the combination of $k_{EP}$ and $d_G$ is a flexible regulation strategy to explain possible implications of bimodal distributions with two NOPs and trimodal

distributions. The peaks are more sensitive to the change of $d_G$ when $k_{EP}$ is small, but to the change of $k_{EP}$ when $d_G$ is large (Figs 4, S2, and S3). S7 Fig also shows different parameters in modulating mRNA distribution. System's parameters would collectively regulate the resulting mRNA distribution, and the parameters related to E-P communication shows a consistent pattern of modulating mRNA distribution across parameter changes. These sensitivity phenomenon provide insights into complex mechanisms of biological processes and are essential for cellular decision-making and environmental adaptation. Third, our mathematical model shows well predictions on the experimental mESCs data, including the mRNA distributions and the relationship between contact probability and mean mRNA level (and CV) under different E-P interactions. The inference results indicate that the slow chromatin dynamics plays an important role in regulating the distribution of mRNAs (S5 Fig). And in particularly, for the slow dynamics, the regulation of upstream to downstream is inseparable, which may cause the gene expression levels of the slow limit are not linear dependence on E-P contacts and eventually lead to the nonlinearity observed in the experiment between transcriptional levels and E-P contact probabilities. These results provide a possible mechanism for E-P interaction translated into gene-expression dynamics.

Our model has several advantages compared to previous work that studied the effect of chromatin structure on gene expression. Xiao *et al.* [31] proposed a mechanism of how E-P signals were involved in the accumulation and removal of the transcription factors favoring transcription to explain the hypersensitivity of transcription to changes in contact frequency. Zuin *et al.* [10] used the E-P encounter probability obtained from the data analysis as the regulator of the upstream to the downstream and assumed that the OFF to ON process was a multi-step process with cumulative effects. However, we note that the corresponding models did not use dynamically fluctuating E-P spatial distances to regulate downstream transcription processes. In our model, random movement of the enhancer and promoter causes fluctuations in E-P spatial distance and regulates gene transcription in real time. In fact, this direct and dynamic regulation leads to the variable rate from OFF to ON, which is practically an alternative to the multi-step process described above. We point out that in the case of fast chromatin dynamics, upstream-to-downstream regulation is performed through the expectation of the E-P spatial distance, which can be equated, in a general way, to the contact probability in Ref. [10]. However, the slow chromatin dynamics also plays an important role in regulating the distribution of mRNAs (see S1 Text). Moreover, our model has the ability to infer the gene-expression dynamics using less information that can be directly measured and leading to better fitting on mESC data to some extent (S4 Fig), especially for the E-P genomic distance in different cell lines, rather than the E-P encounter probability.

Significantly, our modeling framework can also be extended to more complex situations. For example, some experimental studies reported that gene products could affect chromatin structure [66,67]. We can incorporate this feedback into our model by modifying the gene-state-dependent drift function $V(r,s;t)$ in Eq [4]. In addition, our modeling of chromatin motion is not limited to one-pair E-P interaction. The potential extensions include the cases of multiple enhancers to one promoter [68], one enhancer to multiple promoters [6], or super-enhancers [69]. Finally, using multistate models of gene expression to extract insights from enormous experimental data and complex biological phenomena is impressive. Our two-state model is not the default option, and we may adjust the form of the downstream gene expression model to include more complex biological processes such as mRNA splices [70] and cell cycle [71]. However, we still need to balance the complexity and solvability of the model.

Finally, we point out that our theoretical model, which aims to develop a general modeling framework to study 4D gene-expression kinetics, may provide an opportunity for a dialogue

between theoretical studies and biological experiments. We envision that our modeling framework will be helpful for the biophysical analysis of broader in *vivo* cellular processes.

## Supporting information

**S1 Text. It consists of two parts: (1) model description and simulation; (2) fitting experimental data using model.** In the first section, we supplement the details of the model and give the simulation algorithm as well as the statistics analyses of the simulation data. In the second section, we present the procedure for fitting experimental data using our model and comparing the experimental data with theoretical results.
(PDF)

**S1 Fig. Bimodal and trimodal cases under different parameters.** (**A**) The bimodal distribution (with a origin peak and a non-origin peak) of gene expression where $k_{EP} = 0.5$, $\gamma = 1$, $\alpha_{max} = 0.10$, $\alpha_{min} = 0.01$, $\beta = 0.03$, $\mu_{max} = 5$, $\mu_{min} = 2$, $\delta = 0.1$. (**B**) The bimodal distribution (with two origin peaks) of gene expression where $k_{EP} = 0.20$, $\gamma = 100$, $\alpha_{max} = 5$, $\alpha_{min} = 0.30$, $\beta = 0.50$, $\mu_{max} = 8$, $\mu_{min} = 2$, $\delta = 0.1$. (**C**) The trimodal distribution of gene expression where $k_{EP} = 0.1$, $\gamma = 300$, $\alpha_{max} = 0.5$, $\alpha_{min} = 0.05$, $\beta = 0.02$, $\mu_{max} = 10$, $\mu_{min} = 2$, $\delta = 0.1$.
(TIF)

**S2 Fig. E-P interaction can induce bimodal mRNA distribution with two non-origin peaks.** (**A**) Effects of E-P interaction on mRNA distribution. The black lines stand for the boundaries of different peak numbers. The plain is divided into distinctive regions representing unimodal (U) and bimodal (B). The heatmap is the bimodal coefficient. (**B**) Dependence of the most probable mRNA numbers on $k_{EP}$ where $d_G = 60$. The color regions represent different peak numbers. The color bar represents the peak probabilities. (**C**) The example of unimodal/bimodal mRNA distribution. NOP (non-origin peak). The solid lines represent theoretical results and histograms represent numerical results. Parameter values are set as $d_G = 60$, $\alpha_{max} = 5$, $\alpha_{min} = 0.3$, $\beta = 0.5$, $\mu_{max} = 8$, $\mu_{min} = 2$, $\delta = 0.1$, $\gamma = 50$.
(TIF)

**S3 Fig. E-P interaction can induce bimodal mRNA distribution with one origin peak and one non-origin peak.** (**A**) Effects of E-P interaction on mRNA distribution. The black lines stand for the boundaries of different peak numbers. The plain is divided into distinctive regions representing unimodal (U) and bimodal (B). The heatmap is the bimodal coefficient. (**B**) Dependence of the most probable mRNA numbers on $k_{EP}$ where $d_G = 60$. The color regions represent different peak numbers. The color bar represents the peak probabilities. (**C**) The example of unimodal/bimodal mRNA distribution. OP (origin peak), NOP (non-origin peak). The solid lines represent theoretical results and histograms represent numerical results. Parameter values are set as $d_G = 60$, $\alpha_{max} = 0.10$, $\alpha_{min} = 0.01$, $\beta = 0.03$, $\mu_{max} = 5$, $\mu_{min} = 2$, $\delta = 0.1$, $\gamma = 1$.
(TIF)

**S4 Fig. Comparisons the fitting results of our model and Zuin's model to experimental data.** (**A**) KS distances of different cell lines (the KS distance is defined in Eq [16] in the S1 Text). The blue dots and shadow line stand for the KS distance between our model and experimental data, and the dashed blue line represents the mean KS distance for all cell lines. The red shows the KS distance between Zuin's model and experimental data. (**B**) Distribution of mRNA numbers of smRNA-FISH in cell lines $C_6$. The blue line shows the best fit of our model whereas the red line shows the best fit of Zuin's model to the experimental data shown in the histograms. (**C**) CDF of the distributions in (B). The $KS_6^W = 0.0284$ and $KS_6^Z = 0.12$. The

small arrows represent the number of mRNAs corresponding to the KS distance.
(TIF)

**S5 Fig. The contribution of slow chromatin dynamics to data fitting. (A)** The weight of slow chromatin dynamics in fitting distribution ($1/(1+\omega)$). **(B)** KS distances of different cell lines. The blue dots and shadow line stand for the KS distance between the fast distribution and experimental data, and the red shows the KS distance between interpolation distribution and experimental data. The dashed lines represent the mean KS distance for all cell lines. **(C)** The distribution of Cell line 2. **(D)** The distribution of Cell line 6.
(TIF)

**S6 Fig. Comparing the fitting results of model's Hill and linear dependence to experimental data. (A)** KS distances of different cell lines. The red (/green) dots and shadow line stand for the KS distance between Hill (/linear) (Eq [3]/Eq [4] in S1 Text) dependence and experimental data. The blue shows the KS distance between Zuin's model and experimental data. the dashed lines represent the mean KS distance for all cell lines. **(B)** The relationship between $d_{G}$ and contact probability. The blue circles represent the experimental data from the ectopic Sox 2 transgene. The solid red (/green) line is obtained based on the Hill (/linear) dependence. The inset shows the log-log plot. **(C)** Mean eGFP mRNA level plotted against contact probability between the ectopic Sox2 promoter and SCR insertions. The blue polygon shows the experimental data, and the red (/green) polygon shows the corresponding theoretical results based on the Hill (/linear) dependence. **(D)** CV of eGFP level against contact probability. The meanings of symbols are the same as those in (C).
(TIF)

**S7 Fig. Comparisons of parameter's changes on the multimodality of mRNA distribution. (A)** Effects of E-P interaction strengths $k_{EP}$ and friction coefficient $\gamma$ on the pattern of mRNA distribution. Parameters are $d_{G} = 60$, $\alpha_{max} = 5$, $\alpha_{min} = 0.3$, $\beta = 0.5$, $\mu_{max} = 8$, $\mu_{min} = 2$, $\delta = 0.1$. **(B)** Effects of $k_{EP}$ and the ratio of $\mu_{max}/\mu_{min}$ on the pattern of mRNA distribution. The $\mu_{min} = 0.1, 0.3, 0.5$. Parameters are $\alpha_{max} = 0.10$, $\alpha_{min} = 0.01$. **(C)** Effects of $k_{EP}$ and the ratio of $\alpha_{max}/\alpha_{min}$ on the pattern of mRNA distribution. The $\alpha_{min} = 0.01, 0.04, 0.08$. Parameters are $\mu_{max} = 5$, $\mu_{min} = 2$. Each line divides the entire region into two parts, with U indicating a single peak area and M indicating a multimodal area. Other parameter values are set as $d_{G} = 60$, $\beta = 0.03$, $\delta = 0.1$, $\gamma = 1$. The solid line divides the entire region into two parts, with the part marked U indicating unimodal and the part marked M indicating multimodal (including bimodal and trimodal).
(TIF)

## Author Contributions

**Conceptualization:** Jiajun Zhang.

**Investigation:** Zihao Wang, Songhao Luo, Zhenquan Zhang.

**Methodology:** Zihao Wang, Songhao Luo, Zhenquan Zhang, Jiajun Zhang.

**Project administration:** Jiajun Zhang.

**Supervision:** Jiajun Zhang.

**Visualization:** Zihao Wang, Songhao Luo, Zhenquan Zhang.

**Writing – original draft:** Zihao Wang, Songhao Luo, Zhenquan Zhang, Tianshou Zhou, Jiajun Zhang.

**Writing – review & editing:** Zihao Wang, Songhao Luo, Zhenquan Zhang, Tianshou Zhou, Jiajun Zhang.

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
