## [Decision Letter · Decision Letter 0]

14 Sep 2023

Dear Dr. Zhang,

Thank you very much for submitting your manuscript "4D nucleome equation predicts gene expression controlled by long-range enhancer-promoter interaction" for consideration at PLOS Computational Biology.

As with all papers reviewed by the journal, your manuscript was reviewed by members of the editorial board and by several independent reviewers. In light of the reviews (below this email), we would like to invite the resubmission of a significantly-revised version that takes into account the reviewers' comments.

As you will see from the reports copied below, the reviewers find the modeling framework to be potentially valuable. However, they raise several concerns, in particular with regards to the novelty relative to Ref 10.

We cannot make any decision about publication until we have seen the revised manuscript and your response to the reviewers' comments. Your revised manuscript is also likely to be sent to reviewers for further evaluation.

Sincerely,

Maxwell Wing Libbrecht, Ph.D.

Academic Editor

PLOS Computational Biology

Kiran Patil

Section Editor

PLOS Computational Biology

As you will see from the reports copied below, the reviewers find the modeling framework to be potentially valuable. However, they raise several concerns, in particular with regards to the novelty relative to Ref 10.

Reviewer's Responses to Questions

**Comments to the Authors:**

Reviewer #1: In the manuscript ‘4D nucleosome equation predicts gene expression controlled by long-range enhancer promoter interaction’ Wang et al. proposed a theoretical framework for studying gene expression dynamics as a result of enhancer promoter (E-P) interaction. The fundamental concept of the framework is the description of chromatin dynamics and gene expression dynamics in a single equation, as well as the separation of the timescales for the two types of dynamics. The authors showed that simulations based on this framework produced results consistent with both the general understanding of E-P regulation and specific experimental data. Overall, the modeling framework is rigorous and novel. The manuscript is well-written overall. I have a few comments/suggestions regarding the connection between the models and data, as well as the clarity of the manuscript.

1. The biological interpretation of the ‘monomer’ is unclear. It was introduced generically on Page 5 without a description of its meaning. On Page 7, however, it was indicated that each monomer is one nucleosome. When fitting the model to the mESC data, a monomer became a 5kb region, which is more than 10-fold longer than a region spanned by a nucleosome. The confusing usages of the monomer’s meaning needs to be addressed. While the authors can make the interpretation flexible, doing so would mean that the biological meanings for all parameters of the model will change accordingly. This can be problematic when interpreting the simulation results. In addition, it will be helpful if the authors can use realistic distributions of genomic E-P distances to explain the scales of their structural model.

2. The comparison of multimodality of mRNA distribution from simulations and experimental data seems weak. Out of the three references mentioned in that section on Page 16, none of them showed the relationship between E-P regulation and modality of mRNA distribution. It is therefore unclear whether the simulation results are realistic.

3. I suggest that the authors use their framework to explain the source of the multimodality. Is this a result of the chromatin dynamics which produced multiple structural configurations that are more stable than others? If yes, some analysis of the structural model component will be helpful.

4. Page 3, Line 56. ‘Its’ should be ‘whose’, and ‘chromatin’ is misspelled.

5. Page 14, the last sentence. ‘Decrease’ should be ‘decreases’, and ‘trends’ should be ‘tends’.

Reviewer #2: The paper introduces the 4D nucleome equation as a comprehensive theoretical framework for investigating the impact of long-range enhancer-promoter (E-P) interactions on gene expression dynamics. The framework incorporates upstream chromatin motion, downstream mRNA production, and the temporal connection between these processes. The authors derive analytical mRNA distribution and explore how E-P interactions qualitatively influence the characteristics of mRNA distribution. The 4D nucleome equation provides a valuable modeling framework for studying the influence of chromatin dynamics on gene expression kinetics.

However, it is unclear what specific contributions the authors have made compared to previous work. Although the authors briefly discuss a comparison with the model presented in Ref. 10, most of the discussion is relegated to the Supplementary Information (SI). It appears that the authors' model is identical to the one in Ref. 10 in the fast chromatin dynamics regime. In this limit, the contact probability used in Ref. 10 can be directly replaced with the distance used in Eq. 3 of the SI.

Additionally, the authors introduce expressions for the slow chromatin dynamics limit, which are new. However, they do not discuss the contribution of slow dynamics to the fitting of experimental data. While the authors emphasize that their model fits experimental data better than the expression in Ref. 10, this improvement may be expected since the new model has more parameters. It would be interesting if the authors could provide insight into the obtained value for w in Eq. 12 and its impact on fitting with experimental data. Is the contribution from slow chromatin dynamics important for the observed improvement in fitting?

To justify the publication of the manuscript, the authors need to highlight novel observations that were not reported in Ref. 10. Many of the analytical expressions presented in the manuscript are standard for polymer physics and stochastic gene expression models. While it is commendable that the authors report these expressions and combine the two models, the expressions alone may not present sufficiently interesting results for publication.

Furthermore, one of the most notable findings in Ref. 10 is the non-linear dependence of the activation rate on E-P contacts, as demonstrated in Eq. 3 of the SI. Surprisingly, the authors do not mention this non-linearity in the main text and simply adopt the same assumptions as in Ref. 10. This omission prevents the authors from providing alternative explanations for the underlying molecular mechanisms that give rise to the observed non-linearity.

Lastly, there are some minor comments that the authors should address.

1) Firstly, if Eq. 9 in the main text has been derived previously, the authors should provide appropriate citations or clarify the novelty of the expression.

2) Secondly, for the fitting presented in Fig. 5, it would be useful to list the parameters in the SI.

3) Lastly, for Fig. S4, the authors should refer to the SI text that provides the definition of the KS distances.

Addressing these points will strengthen the manuscript and enhance its potential for publication.

Reviewer #3: The authors developed a modeling framework to couple enhancer-promoter interaction with gene expression at different timescales and applied the model to study how the enhancer-promoter interaction affects gene expression dynamics. It considered upstream chromatin motion on a fast timescale and downstream mRNA production on a slow timescale. The authors derived analytical mRNA distribution by following timescale separation method and also numerically solved the equation. They showed that E-P interaction could give rise to multiple shapes of mRNA distributions including bimodal and trimodal distributions. A power-law scaling of gene expression levels in the E-P genomic distance was also demonstrated through both theoretical analysis and numerical simulations. Analysis of experimental data showed consistent results as predicted by the model in mRNA distribution and E-P contact probability under different E-P interactions. The manuscript was clearly written to describe the modeling framework, main results and conclusions, except some grammar mistakes. I would suggest publishing this work if the following comments can be addressed:

On page 6, in the equation F(r,s;t)=V(r,s:t)-∇_r∙(Dlogp(r;t)), it should be ∇_r (Dlogp(r;t)) instead of ∇_r∙(Dlogp(r;t)).

On page 8, please explain why overdamped Langevin equation was applied to model the chromatin dynamics in more detail.

On page 10, when minimizing the total cross entropy function Eq. [6], was it possible to obtain multiple local minimizers? In that case, how were the parameter values determined? Will conclusions still be valid for different parameter values?

On page 13 line 265, V(r)=[(k_NN/d_G +k_EP ) d_s ]/bγ should be V(r)=[(k_NN/d_G +k_EP ) d_s ]/(bγ)?

On page 15 line 318, the last one of the five modes should be unimodal with NOP.

In Fig. S2, the titles of subfigures should be U(1 NOP).

On page 16 line 328-331, the authors discussed different scenarios that the evolution of the mRNA distribution may undergo. I would suggest to add more detailed discussion on the critical conditions or sensitive parameters giving rise to different scenarios and what biological insights can be obtained. What are the different conditions giving rise to results presented in S2 and S3?

On page 17 line 355, different values of d_G were chosen by fitting the experimental data. This information should be provided to clarify how those values were obtained.

Grammar mistakes:

Page 10 line 205, mothod -> method

Page 11 line 223, challenge-> challenging

Page 11 line 236, or a a frozen E-P -> or a frozen E-P

Page 13 line 260, given vs -> given as

Page 14 line 285, CV downregulated by -> CV is downregulated by

Page 14 line 297, the mRNA level decrease and trends -> the mRNA level decreases and tends

Page 16 line 329, it should be ‘still changes from unimodal to the bimodal and then to unimodal again’.

**Have the authors made all data and (if applicable) computational code underlying the findings in their manuscript fully available?**

Reviewer #1: Yes

Reviewer #2: None

Reviewer #3: Yes

PLOS authors have the option to publish the peer review history of their article (what does this mean?). If published, this will include your full peer review and any attached files.

Reviewer #1: No

Reviewer #2: No

Reviewer #3: No
---

## [Decision Letter · Decision Letter 1]

28 Nov 2023

Dear Dr. Zhang,

We are pleased to inform you that your manuscript '4D nucleome equation predicts gene expression controlled by long-range enhancer-promoter interaction' has been provisionally accepted for publication in PLOS Computational Biology.

Best regards,

Maxwell Wing Libbrecht, Ph.D.

Academic Editor

PLOS Computational Biology

Kiran Patil

Section Editor

PLOS Computational Biology

Reviewer's Responses to Questions

**Comments to the Authors:**

Reviewer #1: The authors have addressed all my comments adequately.

Reviewer #2: The authors have successfully addressed all my concerns.

**Have the authors made all data and (if applicable) computational code underlying the findings in their manuscript fully available?**

Reviewer #1: Yes

Reviewer #2: None

PLOS authors have the option to publish the peer review history of their article (what does this mean?). If published, this will include your full peer review and any attached files.

Reviewer #1: No

Reviewer #2: No

---

## [Editor Report · Acceptance letter]

11 Dec 2023

PCOMPBIOL-D-23-00602R1 

4D nucleome equation predicts gene expression controlled by long-range enhancer-promoter interaction

Dear Dr Zhang,

I am pleased to inform you that your manuscript has been formally accepted for publication in PLOS Computational Biology. Your manuscript is now with our production department and you will be notified of the publication date in due course.

With kind regards,

Anita Estes
